# Melatonin-Mediated Colonic Microbiota Metabolite Butyrate Prevents Acute Sleep Deprivation-Induced Colitis in Mice

**DOI:** 10.3390/ijms222111894

**Published:** 2021-11-02

**Authors:** Ting Gao, Zixu Wang, Yulan Dong, Jing Cao, Yaoxing Chen

**Affiliations:** 1Laboratory of Anatomy of Domestic Animals, College of Veterinary Medicine, China Agricultural University, Haidian, Beijing 100193, China; gaotinginging@163.com (T.G.); zxwang@cau.edu.cn (Z.W.); ylbcdong@cau.edu.cn (Y.D.); caojing@cau.edu.cn (J.C.); 2Department of Nutrition and Health, China Agricultural University, Haidian, Beijing 100193, China

**Keywords:** melatonin, sleep deprivation, colitis, gut microbiota, butyrate

## Abstract

Radical cure colitis is a severe public health threat worldwide. Our previous studies have confirmed that melatonin can effectively improve gut microbiota disorder and mucosal injury caused by sleep deprivation (SD). The present study further explored the mechanism whereby exogenous melatonin prevented SD-induced colitis. 16S rRNA high-throughput sequencing and metabolomics analysis were used to explore the correlation between SD-induced colitis and intestinal microbiota and metabolite composition in mice. Fecal microbiota transplantation (FMT) and melatonin or butyrate supplementation tests verified the core role of gut microbiota in melatonin-alleviating SD-induced colitis. Further, in vitro tests studied the modulatory mechanism of metabolite butyrate. The results demonstrated that SD leads to reductions in plasma melatonin levels and colonic Card9 expression and consequent occurrence of colitis and gut microbiota disorder, especially the downregulation of *Faecalibacterium* and butyrate levels. The FMT from SD-mice to normal mice could restore SD-like colitis, while butyrate supplementation to SD-mice inhibited the occurrence of colitis, but with no change in the plasma melatonin level in both treatments. However, melatonin supplementation reversed all inductions in SD-mice. In intestinal epithelial cells, the inflammatory ameliorative effect of butyrate was blocked with pretreatments of HDAC3 agonist and HIF-1α antagonist but was mimicked by GSK-3β and p-P65 antagonists. Therefore, the administration of MLT may be a better therapy for SD-induced colitis relative to butyrate. A feasible mechanism would involve that melatonin up-regulated the *Faecalibacterium* population and production of its metabolite butyrate and MCT1 expression and inhibited HDAC3 in the colon, which would allow p-GSK-3β/β-catenin/HIF-1α activation and NF-κB/NLRP3 suppression to up-regulate Card9 expression and suppress inflammation response.

## 1. Introduction

Inflammatory bowel disease (IBD), which is an umbrella term for ulcerative colitis (UC) and Crohn’s disease (CD), is characterized by chronic and recurrent inflammation in the gut. Its incidence and prevalence have increased worldwide in the last 50 years and IBD affects about 3 million people in the United States [1]. To radically cure IBD is a considerable challenge because IBD is defined as a multifactorial disease, and its pathogenesis is still unclear. IBD is caused by a complex interplay between genetic, immunologic, microbial, and environmental factors, making the development of a subtype-specific treatment challenging. Thus, efforts are ongoing to develop personalized therapies for the various types of IBD and improve the quality of life of patients [2,3,4]. Our previous studies have demonstrated that melatonin (MLT) can effectively improve sleep deprivation (SD)-induced gut microbiota disturbances and intestinal mucosal damage [5]. Sleep impairment in IBD patients is a common problem that deserves attention in everyday clinical practice [6]. In addition, sleep is also closely related to many disorders including depression, etc. [7]. Psychological stress causes an increase in intestinal permeability and destruction of the intestinal mucosal barrier via altering gut microbiota [8]. Moreover, accumulating evidence suggests that host intestinal barrier maintenance senses the gut bacteria not only directly through recognition of the intestinal epithelium [9], but also by sensing microbial metabolites, which influence the host immune response in the gut and beyond [10,11]. However, it is unclear how changes in gut microbiota metabolites when the host is exposed to SD, and whether microbiota metabolites mediate the effect of MLT on relieving intestinal mucosal damage induced by SD.

Research has revealed a group of molecules known as short-chain fatty acids (SCFAs), which are involved in the cross-talk between the microbiome, the epithelium, and the immune system [12]. As intestinal microbiota metabolites, SCFAs have been shown to maintain intestinal homeostasis by protecting epithelial barrier integrity [13,14], promoting sIgA production [15], and regulating T-cell differentiation [16,17]. Particularly, SCFAs are proposed to have anti-inflammatory activities in the colonic mucosa [18], and the intrarectal delivery of SCFAs have been previously shown to prevent colitis [19]. SCFAs may permeate the cell membrane using two different solute transporters, namely the proton-coupled monocarboxylate-transporter 1 (MCT1/SLC16A1) and the sodium-coupled monocarboxylatetransporter 1 (SMCT1/SLC5A8) [20,21]. Alternatively, SCFAs may activate signaling pathways via at least three different GPCRs: GPR41, GPR43, and GPR109 [22,23,24]. Moreover, SCFAs can regulate cell functions via histone deacetylase (HDAC) inhibition [16,25,26]. Studies have shown that the transporters are the main transport routes for butyrate and these are abundantly expressed in the colon [27]. Butyrate, a member of the SCFAs, is mostly produced by *Faecalibacterium* and serves as the principal source of energy for colonocytes; it exerts multiple effects on a variety of cell types leading to immune modulation, cell cycle inhibition, induction of programmed cell death, and cellular differentiation [27,28,29]. Although butyrate may impact physiological functions and homeostasis to maintain health, whether butyrate mediates MLT in relieving acute SD-induced colitis and the underlying mechanism remain unclear.

Based on our previous study, which showed that MLT can effectively relieve acute SD-induced colon inflammation and gut microbiota disorders, the present study further explored the effect of gut microbiota metabolite butyrate on the improvement of SD-induced colitis by MLT. The present study was conducted to (1) investigate the effect of SD on intestinal microbiota and metabolites composition, including SCFAs in mice; (2) further verify the core role of intestinal microbiota and metabolites disorder in SD-induced colitis using antibiotics and FMT; (3) supplementing MLT or butyrate to SD mice confirm MLT-mediated butyrate improve the colitis induced by SD, and (4) explore the signaling pathway in MLT-mediated butyrate relieve of acute SD-induced colitis using in vivo and in vitro tests.

## 2. Materials and Methods

All experiments were conducted according to the Guide for the Care and Use of Laboratory Animals published by the Animal Welfare Committee of the Agricultural Research Organization, China Agricultural University (Approval No. CAU20170911-2).

### 2.1. Cell Culture and Treatment

Mouse normal colon intestinal epithelial cells (BALB-5047, USA) were obtained from the RIKEN Cell Bank (Ibaraki, Japan). The cells were cultured in 96-well culture plates (5 × 10^6^ cells/mL) and 12-well culture plates (5 × 10^5^ cells/mL). Some LPS-treated cells (10 nM, Solarbio Ltd., Beijing, China) were treated with 2 μM TWS119 (a selective GSK-3β antagonist; MCE, Nanjing, USA; LPS + TWS-cells), 100 μM PDTC (an NF-κB antagonist; MCE, Nanjing, USA; LPS + PDTC-cells), or 5 mM butyrate (Sigma-Aldrich, St. Louis, MO, USA; LPS + Butyrate cells). After butyrate supplementation for 30 min, some LPS + Butyrate cells were sequentially treated with 50 μM ITSA-1 (a nonselective HDAC3 agonist; MCE, New Jersey, USA; LPS + Butyrate + ITSA-1-cells) or 5 nM Chetomin (a nonselective HIF-1α antagonist; Abcam, Cambridge, MA, USA; LPS + Butyrate + Chetomin-cells). Each plate of treated cells was incubated for 24 h.

The cells from the 96-well culture plates were assessed for proliferation activity using an MTT (3-(4,5)-dimethylthiahiazo (-z-y1)-3,5-di-phenytetrazoliumromide; Sigma, St. Louis, MO, USA) assay. The optical density was determined using a microplate reader (Model 680, Bio-Rad, St. Louis, MO, USA) equipped with a 570 nm wavelength filter. The cells from the 12-well culture plates were collected for LDH assessment and Western blotting. Each assay used a repeat of 8 wells.

### 2.2. LDH Assessment

The cell supernatants were detected using an LDH assay kit (Solarbio Ltd., Beijing, China) according to the manufacturer’s instructions. The LDH data were measured at 450 nm wavelength using a microplate reader (Model 680, Bio-Rad, St. Louis, MO, USA) and were expressed as U/10^4^ cells. Each sample was assayed three times.

### 2.3. Animal Model Establishment

A total of 132 male SPF ICR mice (8 weeks old; Vital River Laboratory Animal Technology Co. Ltd., Beijing, China) were housed in 22 cages (6 mice/cage) under conventional conditions. After acclimatization for one week, the mice were randomly divided into eleven groups: sleep deprivation (SD), SD + antibiotics (SD + ABs), SD + melatonin supplementation (SD + MLT), SD + antibiotics + melatonin supplementation (SD + Abs + MLT), SD + antibiotics + butyrate supplementation (SD + Abs + butyrate), only antibiotic supplementation (ABs), non-sleep-deprived control (CON) groups, fecal microbiota transplantation (FMT) from mice of CON group (F-CON), FMT from mice of SD group (F-SD), FMT from mice of SD + MLT group (F-SM) and FMT from mice of vehicle (F-R). For more details see the Appendix A.

### 2.4. Gut Microbiota Analysis

The colonic contents were collected and stored at −80 °C. Bacterial genomic DNA was extracted from frozen intestinal contents using the QIAamp DNA Stool Mini Kit from Qiagen (Hilden, Germany) according to the manufacturer’s instructions. The 16S rRNA gene, comprising V3 and V4 regions, was amplified using PCR and composite specific bacterial primers as described previously [5]. All effective reads from each sample were clustered into operational taxonomic units based on a 97% sequence similarity according to UCLUST.

### 2.5. Metabolomics Profiling

We performed LC-MS analyses on a quadrupole-time-of-flight (Q-TOF) 6510 mass spectrometer (Agilent Technologies, Santa Clara, CA, USA) with an electrospray ionization source. For more details see the Appendix A.

### 2.6. Faecal Occult Blood Test

A small quantity of faces was applied to the center of the slide, and three drops of 10 g/L methylaminophenol sulphate solution and three drops of 3% hydrogen peroxide solution were added dropwise. The results were observed and images were taken immediately. The judging criterion was negative: (−) No rosy red or cherry red after 3 min; Positive: (+) Rose red or cherry red appears within 30–60 s; strong positive: (++) Rose red or cherry red appears immediately; strongest positive: (+++) Deep rose red or deep cherry red appears immediately.

### 2.7. Intestinal Permeability to FITC-Dextran

Before the experiment ended at 6:00 am, all mice were deprived of food for 2 h and orally gavaged with 0.6 mg/g body weight 4 kDa fluorescein isothiocyanate (FITC)-dextran at a concentration of 80 mg/mL 1 h before euthanasia. Blood (approximately 1 mL) was collected by retro-orbital eye bleed and centrifuged (500× *g*, 10 min) to collect serum (around 500 µL). The fluorescence in the serum was measured using a fluorescent spectrophotometer with 485 nm excitation and 535 nm emission. A standard curve was created by diluting FITC-dextran in PBS. The concentration of FITC-dextran in the serum was calculated using the standard curve.

### 2.8. Enzyme-Linked Immunosorbent Assay (ELISA)

Plasma samples were collected for the detection of MLT, and colon samples were collected for the detection of inflammatory factors (IL-10, IFN-γ, IL-1β, IL-6, TNF-α, and IL-17) using a competitive ELISA assay (Uscn Life Science, Inc., Wuhan, China). All tests were performed according to the manufacturer’s instructions. Eight samples were used in each group. Each sample was tested in triplicate. The intra-assay coefficient of variation (CV) was <10%, and the inter-assay CV was <12%. The data were measured using a microplate reader (Model 680, Bio-Rad, St. Louis, MO, USA) equipped with a 450 nm filter. The data were expressed as pg/mL for the plasma MLT level and pg/mg protein for the IL-10, IFN-γ, IL-1β, IL-6, TNF-α, and IL-17 levels of the colonic tissue.

### 2.9. Histological Staining

Forty-two colonic segments were immediately fixed in 4% paraformaldehyde in 0.1 M phosphate-buffered saline (pH 7.4, 4 °C) for 48 h and embedded in paraffin. Tissue cross-sections (5 μm) were stained with haematoxylin and eosin (H&E). At least 30 random fields in six sections of each sample were photographed at × 400 magnification with a microscope (BX51; Olympus, Tokyo, Japan), and a total of at least 180 fields were analyzed per treatment. The scoring criterion was: (a) 0: No evidence of inflammation; (b) 2: Low level of inflammation, with scattered infiltrating mononuclear cells (1–2 foci); (c) 4: Moderate inflammation with multiple foci; (d) 6: High level of inflammation, with increased vascular density and marked wall thickening; (e) 8: Maximal severity of inflammation, with transmural leukocyte infiltration.

### 2.10. Immunohistochemical Staining

Paraffin sections were incubated overnight at 4 °C with a rabbit anti-mouse monoclonal primary antibody (Card9, 1:500; Abcam, Cambridge, MA, USA). For more details see the Appendix A.

### 2.11. Western Blotting

Total protein (obtained from colon samples and IECs) was extracted using lysis buffer (62.5 mmol/L Tris-HCl, 2% SDS, and 10% glycerol; pH 6.8). For more details see the Appendix A.

### 2.12. Statistical Analysis of Data

Data were analyzed using SPSS 10.0 statistical software (SPSS, Inc., Chicago, IL, USA). Data is first analyzed for normality. Data conforming to normality analysis are expressed as the mean ± SD. Differences between groups were statistically analyzed using multi-way ANOVA followed by one-way ANOVA analysis, which was used to determine the significance of differences among groups and denoted as follows: different lowercase letters or different uppercase letters: *p* < 0.05; different letters and one contains uppercase letters and the other is lowercase letters: *p* < 0.01; same letter: *p* > 0.05.

## 3. Results

### 3.1. MLT Improved the Gut Microbiota Population in SD Mice

To investigate the improvement effect of MLT on SD-induced gut microbiota disorder, we compared the changes in colonic microbiota components among the CON, SD, and SD + MLT groups. High-throughput pyrosequencing of 16S rRNA showed the distribution of colonic microbiota. Co-inertia analysis showed dysregulation of intestinal microbiota in SD mice. Compared with the CON group, the proportion of *Firmicutes* and *Proteobacteria* in SD mice increased significantly by 21.9 ± 0.12% (*p* = 0.002) and 39.8 ± 0.026% (*p* = 0.007), while the proportion of *Bacteroidetes* and *Verrucomicrobia* decreased significantly by 30.5 ± 0.032% (*p* = 0.022) and 59.8 ± 0.006% (*p* = 0.001), respectively, (Figure 1A,B). In particular, the content of colonic *Faecalibacterium* reduced significantly by 60.3 ± 0.034% (*p* < 0.001) in the SD group compared with that in the CON group (Figure 2D). However, supplementation with MLT restored these changes in flora populations and resulted in no significant difference between the CON and SD + MLT groups (*p* > 0.209; Figure 1C). Meanwhile, we correlated the most remarkable changes in intestinal microbes and metabolites of the three groups using correlation network analysis (Figure 1D–F, Appendix A). After SD, the levels of metabolites associated with *Firmicutes* and *Proteobacteria* were significantly upregulated by 15.4 ± 0.029% (*p* = 0.003) and 27.9 ± 0.065% (*p* = 0.013), respectively, compared with those in the CON group, but were downregulated in *Bacteroidetes* by 15.9 ± 0.018% (*p* = 0.032). Similarly, this process was dramatically suppressed by MLT supplementation. As shown in Figure 1G, the most prominent alteration associated with intestinal microbiota and its metabolites involved 12 metabolites and 83 intestinal microbiota. The content of butyrate was directly proportional to the abundance of *Faecalibactium*, *Lachnospiraceae*, *Erysipelatoclostridium*, and *Butyricicoccus*, but was inversely proportional to the abundance of *Alistipes*, *Mollicutes*, and *Rikenellaceae*.

### 3.2. MLT Improved Gut Microbiota Metabolite Disproportionality in SD Mice

We further analyzed the changes in metabolites that responded to the exposure of SD mice to MLT. The analysis of the metabolites showed that there were 1023 metabolites in the colon. The β-diversity analysis showed an obvious clustering of microbiota metabolite composition of all CON, SD, and SD+MLT groups (Figure 3A). SD led to a significant increase in colonic microbiota metabolite dispersion, which suggested a decrease in microbiota metabolite homogeneity. The Venn diagram indicated that different treatments resulted in different metabolite changes (Figure 3B). Compared with the CON group, 51 metabolite contents were changed in the SD group, while MLT supplementation restored the content of 27 of these metabolites. The 27 metabolites included butyrate, L-Proline, and Deoxyinosine, which are closely related to treatments such as SD and MLT supplementation. We then performed a statistical analysis of the differences in metabolite expression levels among the CON, SD, and SD + MLT groups (Figure 3C,D and Appendix A). Specifically, compared with the CON group, 25 metabolites were upregulated and 26 were downregulated after SD, while MLT supplementation reversed these processes (Figure 3C). After MLT supplementation, there were 103 metabolites increased and 28 metabolites decreased, compared with the SD group. Furthermore, we screened the 49 most changed metabolites in the three groups (Figure 3E). Of these, 19 metabolites were significantly reduced and 17 metabolites increased in the SD group compared with those in the CON group. Figure 3F–K summarizes the six metabolites with the most significant changes caused by SD. There was a remarkable decrease in the content of butyrate (75.9 ± 0.039%, *p* = 0.007, Figure 3F), acetylcarnitine (41.7 ± 0.045%, *p* = 0.009, Figure 3G), deoxyadenosine (45.7 ± 0.051%, *p* = 0.002, Figure 3H), and acetylcholine (46.4 ± 0.021%, *p* < 0.001, Figure 3I) as well as an increase in the content of L-proline (108.1 ± 0.09%, *p* < 0.001, Figure 3H) and dopamine (114.2 ± 0.010%, *p* = 0.001, Figure 3K) in the SD group compared to those in the CON group. However, MLT supplementation reversed these changes to the control levels (*p* > 0.55), indicating there was no significant difference between the control group and the MLT supplementation group.

To identify how the changes in intestinal microbiota metabolites affect host signaling pathways, we classified the annotation results of differential metabolites according to the pathway types in KEGG (Figure 2A,B and Appendix A). SD mainly affects the following signaling pathways: metabolic pathways, cAMP-PKG signaling pathway, neuroactive ligand-receptor interaction, gap junction, cocaine addiction, bile secretion, tight junction, Parkinson’s disease, and FOXO signaling pathway. Meanwhile, the SD-induced pathway changes were closely linked to the production of reactive oxygen species, tight junction proteins, inflammatory factors, HDAC, and the NF-κB pathway (Figure 2C). It can be speculated that SD leads to changes in metabolites, especially a decrease in butyrate, which in turn affects these pathways and functions, ultimately resulting in colitis.

### 3.3. FMT Promote Re-Establishment of the Intestinal Microecology in Mice

As illustrated in Figure 4, FMT from SD mice induced a colitis, inflammatory response and intestinal microbiota dysbiosis. Compared with the F-CON group, body weight and colonic length decreased by 4.3 ± 0.021% (*p* = 0.026, Figure 4A) and 54.3 ± 0.027% (*p* = 0.004, Figure 4D,E), while colonic permeability and histopathological score increased by 38.1 ± 0.003% (*p* = 0.002, Figure 4B,C) and 51.5 ± 0.038% (*p* < 0.001, Figure 4F,G) in the F-SD group. However, the stimulatory effects of F-SD on changes in colitis were reversed in the colon by F-SM supplementation.

Further, we examined whether F-SD caused colonic inflammation and intestinal microbiota imbalance. The IL-10 and IFN-γ levels were significantly reduced by 36.3 ± 0.014% (*p* = 0.010, Figure 4H) and 41.4 ± 0.041% (*p* = 0.002, Figure 4I) in the F-SD group, respectively, versus the F-CON group. By contrast, IL-1β and IL-6 content was significantly elevated by 32.5 ± 0.005% (*p* = 0.032, Figure 4J) and 46.3 ± 0.049% (*p* = 0.010, Figure 4K) in the colon of the F-SD group versus the control group. However, in the F-SM group, all indexes were restored to the level of the F-CON group, resulting in no statistically significant differences between the control group and the F-SM and F-MT groups (*p* > 0.783).

Moreover, high-throughput pyrosequencing of 16S rRNA demonstrated that the relative abundance of Firmicutes and Proteobacteria increased significantly and the content of Bacteroidetes and *Faecalibacterium* decreased markedly in the F-SD-treated mice compared with the F-CON group. However, in the F-SM group, all indexes were restored to the level of the F-CON group (Appendix A).

Collectively, these results indicated that FMT promotes the re-establishment of the intestinal microecology.

### 3.4. MLT Supplementation Improved Plasma MLT Levels and Colonic Butyrate Production in SD Mice

As shown in Table 1, compared with the CON group, plasma MLT levels decreased by 62.6 ± 0.037% (*p* = 0.026) and 62.2 ± 0.004% (*p* = 0.040) in the SD and SD + ABs groups, but no change was observed in the ABs group. However, after MLT supplementation, the MLT content increased by 73.6 ± 0.004–117.8 ± 0.007% (*p* = 0.026–0.040) and 72.7 ± 0.014–141.5 ± 0.071% (*p* = 0.015–0.024) in the SD + MLT and SD + Abs + MLT groups, respectively, relative to that in the SD and SD + ABs groups, but there were no significant differences among that in the CON, SD + MLT, and SD + Abs + MLT groups. In contrast, butyrate supplementation did not alter the MLT level in the SD + Abs + Butyrate group compared to that in the SD + ABs group (*p* = 0.206).

Similar to the change in plasma MLT levels, the colonic butyrate content decreased markedly in the SD (20.4 ± 0.079%, *p* = 0.003), ABs (20.5 ± 0.009%, *p* = 0.002), and SD + ABs (21.9 ± 0.035%, *p* = 0.002) groups compared with that in the CON group, whereas it increased in the SD + MLT (22.8 ± 0.052%, *p* = 0.012), SD + Abs + MLT (23.7 ± 0.058%, *p* = 0.009), and SD + Abs + Butyrate (42.3 ± 0.064%, *p* = 0.002) groups compared with that in the SD group. However, there were no significant differences in the butyrate content among the SD, ABs, and SD + ABs groups (*p* > 0.126) or among the CON, SD + MLT, SD + Abs + MLT, and SD + Abs + Butyrate groups (*p* > 0.626). In contrast to butyrate, colonic acetate and propionate contents did not change in all treatment groups (*p* > 0.734).

### 3.5. Effect of Butyrate on MLT Improved Colitis in SD Mice

To investigate whether butyrate could mediate the MLT-improved SD-induced colitis, we observed changes in the body weight, fecal occult blood, colon length, colonic permeability, histopathological score, release of inflammatory cytokines, and expression of Card9 gene in the colon among the CON, SD, ABs, SD + ABs, SD + MLT, SD + Abs + MLT, and SD + Abs + Butyrate groups. Compared with the CON group, we found an occurrence of colitis along with a decrease in MLT in the SD and SD + ABs groups group and butyrate in the SD, ABs, and SD + ABs groups (Table 1), which showed a more serious fecal occult blood (Figure 5B), a decrease in body weight (4.8 ± 0.046–7.9 ± 0.025%, *p* = 0.032–0.045; Figure 5A) and colon length (21.9 ± 0.007–29.7 ± 0.009%, *p* = 0.000–0.001; Figure 5D,E), a downregulation of IL-10 (20.1 ± 0.018–30.7 ± 0.041%, *p* = 0.000–0.002; Figure 5K), IFN-γ (27.1 ± 0.019–37.9 ± 0.006%, *p* = 0.000–0.001; Figure 5L), and Card9 (28.3 ± 0.008–56.8 ± 0.049%, *p* = 0.003–0.008; Figure 5H–J), an increase in permeability (11.2 ± 0.029–175.6 ± 0.025%, *p* = 0.001–0.004; Figure 5C) and histopathological score (177.0 ± 0.020–192.3 ± 0.054%, *p* = 0.000–0.003; Figure 5F,G), and an upregulation of TNF-α (29.5 ± 0.057–30.2 ± 0.031%, *p* = 0.000–0.005; Figure 5M) and IL-17 (32.6 ± 0.020–35.0 ± 0.032%, *p* = 0.015–0.040; Figure 5N).

In contrast, MLT supplementation reversed the SD-induced changes in colitis and inflammatory response along with an increase in MLT and butyrate; no significant difference was observed in fecal occult blood, body weight (*p* > 0.052), colon length (*p* > 0.099), colonic permeability (*p* > 0.335), histopathological score (*p* > 0.243), anti-inflammatory cytokines (IL-10 and IFN-γ, *p* > 0.283), proinflammatory cytokines (TNF-α and IL-17, *p* > 0.215), and enteritis sensitive protein Card9 (*p* > 0.220) among the SD + MLT, SD + Abs + MLT, and CON groups. Similar to MLT supplementation, butyrate supplementation improved the SD-induced colitis and inflammatory response along with an increase in butyrate levels; no difference was observed in all the parameters between the SD + Abs + Butyrate and CON groups. However, SD + Abs + Butyrate treatment did not alter the reduction in MLT compared to that in the SD + MLT and SD + Abs + MLT treatments.

### 3.6. Effect of MCT1/HDAC3/GSK-3β/HIF-1α/NF-κB Pathway on Butyrate Mediates MLT-Improved Colitis

To investigate the mechanism of butyrate-mediated MLT-improved SD-induced colitis, the signaling pathway was studied using both in vitro and in vivo experiments. In vivo, compared with the CON group, there was an obvious downregulation in the expression of MCT1 (9.1 ± 0.011–57.8 ± 0.062%, *p* = 0.018–0.038; Figure 6A), p-GSK-3β/t-GSK-3β (30.0 ± 0.033–38.0 ± 0.002%, *p* = 0.012–0.040; Figure 6C), β-catenin (69.8 ± 0.042–71.2 ± 0.061%, *p* = 0.010–0.030; Figure 6D), and HIF-1α (29.3 ± 0.010–29.5 ± 0.056%, *p* = 0.001–0.008; Figure 6E) as well as a significant upregulation in the expression of HDAC3 (44.5 ± 0.052–47.7 ± 0.019%, *p* = 0.032–0.034; Figure 6B), p-IκB (23.0 ± 0.042–30.2 ± 0.004%, *p* = 0.011–0.017; Figure 6F), p-P65 (71.4 ± 0.001–89.8 ± 0.036%, *p* = 0.012–0.038; Figure 6G), and NLRP3 (82.1 ± 0.029–107.2 ± 0.045%, *p* = 0.000–0.003; Figure 6H) in the SD, ABs, and SD + ABs groups. However, MLT and butyrate supplementation reversed these changes, including a downregulation in HDAC3 (30.6 ± 0.012–45.7 ± 0.016%, *p* = 0.020–0.042), p-IκB (36.1 ± 0.025–46.3 ± 0.034%, *p* = 0.012–0.017), p-P65 (29.4 ± 0.056–48.9 ± 0.041%, *p* = 0.001–0.012), and NLRP3 (21.4 ± 0.026–33.9 ± 0.039%, *p* = 0.001–0.003) as well as an upregulation in MCT1 (25.1 ± 0.065–27.5 ± 0.097%, *p* = 0.011–0.017), p-GSK-3β (24.0 ± 0.009–41.4 ± 0.031%, *p* = 0.030–0.042), β-catenin (13.6 ± 0.078–40.6 ± 0.049%, *p* = 0.012–0.039) and HIF-1α (21.4 ± 0.018–33.9 ± 0.001%, *p* = 0.001) in the SD + MLT, SD + Abs + MLT, and SD + Abs + Butyrate groups relative to those in the SD group, and resulted in no significant difference between the MLT/Butyrate-treated and CON groups (*p* > 0.153).

In vitro, an LPS-induced IEC model with or without butyrate administration was established. LPS induced an upregulation of lactate dehydrogenase (LDH) (56.5 ± 0.014%, *p* = 0.003; Figure 6I), HDAC3 (113.3 ± 0.009%, *p* = 0.019; Figure 7C), p-IκB (18.7 ± 0.032%, *p* = 0.010; Figure 7G), p-P65 (52.0 ± 0.026%, *p* = 0.029; Figure 7H), and NLRP3 (80.2 ± 0.004%, *p* < 0.001; Figure 7I, and a downregulation of Card9 (27.8 ± 0.039%, *p* = 0.008; Figure 7A), MCT1 (60.3 ± 0.069%, *p* = 0.018; Figure 7B), p-GSK-3β/t-GSK-3β (45.1 ± 0.62%, *p* = 0.030; Figure 7D), β-catenin (22.3 ± 0.028%, *p* = 0.006; Figure 7E), and HIF-1α (54.6 ± 0.022%, *p* < 0.001; Figure 7F) compared with the control IECs. The LPS-induced effect was blocked by pre-treatment with TWS119, a GSK-3β antagonist, which showed an upregulation of β-catenin (42.0 ± 0.092%, *p* < 0.001), HIF-1α (63.3 ± 0.005%, *p* = 0.002), and Card9 (76.8 ± 0.044%, *p* < 0.001), and a downregulation of p-IκB (71.8 ± 0.055%, *p* = 0.036), p-P65 (63.1 ± 0.024%, *p* < 0.001), NLRP3 (56.8 ± 0.062%, *p* < 0.001), and LDH (37.9 ± 0.011%, *p* = 0.002) in the LPS + TWS-treated IECs, but no changes were observed in MCT1, HDAC3, and p-GSK-3β proteins (*p* > 0.058). Similarly, pre-treatment with PDTC to suppress p65 phosphorylation blocked the LPS-induced effect and upregulated the expression of Card9 (63.8 ± 0.010%, *p* = 0.004) and caused a downregulation of p-IκB (45.2 ± 0.041%, *p* = 0.032), p-P65 (54.5 ± 0.025%, *p* < 0.001), NLRP3 (44.5 ± 0.001%, *p* = 0.002), and LDH (37.3 ± 0.038%, *p* = 0.002) in the LPS + PDTC-treated IECs, but no changes were observed in MCT1, HDAC3, p-GSK-3β, β-catenin, and HIF-1α proteins (*p* > 0.056).

In contrast, butyrate administration effectively reversed the LPS-induced changes and resulted in the upregulation of Card9 (11.0 ± 0.004%, *p* = 0.010), MCT1 (24.7 ± 0.059%, *p* = 0.018), p-GSK-3β/t-GSK-3β (13.4 ± 0.084%, *p* = 0.030), β-catenin (12.9 ± 0.037%, *p* = 0.006), and HIF-1α (11.0 ± 0.077%, *p* < 0.001), and the downregulation of LDH (30.6 ± 0.012%, *p* = 0.012), HDAC3 (22.4 ± 0.007%, *p* = 0.031), p-IκB (61.7 ± 0.005%, *p* = 0.040), p-P65 (14.8 ± 0.070%, *p* = 0.023), and NLRP3 (25.2 ± 0.027%, *p* = 0.008) compared with those in the LPS-treated IECs. However, there was no significant difference between the LPS + Butyrate-treated and the control IECs (*p* > 0.052). This improving effect of butyrate was blocked by the administration of ITSA-1, an HDAC3 agonist, which led to the downregulation of p-GSK-3β/t-GSK-3β (25.0 ± 0.038%, *p* = 0.012), β-catenin (38.4 ± 0.032%, *p* = 0.003), HIF-1α (13.3 ± 0.045%, *p* < 0.001), and Card9 (83.5 ± 0.009%, *p* = 0.002), and upregulation of p-IκB (33.2 ± 0.043%, *p* = 0.021), p-P65 (14.7 ± 0.012%, *p* = 0.040), NLRP3 (23.8 ± 0.007%, *p* < 0.001), and LDH (59.0 ± 0.001%, *p* < 0.001) in the LPS + Butyrate + ITSA-1-treated IECs, but no changes were observed in MCT1 and HDAC3 proteins (*p* > 0.549). Similarly, administration of Chetomin, an HIF-1α antagonist, countered the improving effect of butyrate, and led to a downregulation of Card9 (10.6 ± 0.038%, *p* = 0.043) and upregulation of p-IκB (23.8 ± 0.083%, *p* = 0.010), p-P65 (55.7 ± 0.045%, *p* = 0.039), NLRP3 (23.4 ± 0.002%, *p* = 0.002), and LDH (58.3 ± 0.034%, *p* < 0.001) in the LPS + Butyrate + Chetomin-treated IECs, but no changes were observed in MCT1, HDAC3, p-GSK-3β, β-catenin, and HIF-1α proteins (*p* > 0.029).

## 4. Discussions

Our previous studies showed that the administration of exogenous MLT can effectively alleviate intestinal barrier dysfunction caused by SD [5,30]. The present study further explored the role of MLT in acute 72 h SD-induced gut microbiota metabolite disproportionality. Consistent with a decrease in plasma MLT levels, we observed an imbalance in microbiota composition in the acute SD-mice, which showed an increase in abundance of Firmicutes and Proteobacteria and a decrease in the abundance of Bacteroidetes and Verrucomicrobia, especially the decline in *Faecalibacterium*. Our results are supported by studies that found an increase in the content of Firmicutes and Proteobacteria, and a decrease in the content of Verrucomicrobia in the faces of obese mice fed a high-fat diet [31]. However, unlike our observation, Zhang found that SD did not have any significant effect on changes in the composition of the gut microbiota, as reported in humans and rats [32]. This discrepancy could be due to differences in model organisms (mice versus humans and rats) and samples for microbiota composition analysis (colonic content versus faces). A previous study demonstrated that the imbalance in the microbial community towards a dysbiotic state is more pronounced in colonic samples than in fecal samples in patients with Crohn’s disease [33], which indicates that the microbiota composition of the colonic contents can better reflect the changes in intestinal microbiota in colitis.

*Faecalibactium* is one of the most abundant butyrate-producing bacteria in the gastrointestinal tract [34]. In this study, 51 metabolite levels were affected in SD mice. Compared with the control group, 25 metabolites increased significantly, and 26 decreased. Of these, butyrate decreased the most. Liquid phase analysis also further demonstrated that the butyrate content decreased significantly in the colon of SD mice, but no changes were observed in acetate and propionate contents. Previous studies have also shown that physiological stressors could lead to a significant decrease in butyrate [35].

Moreover, the mice, receiving feces microbiota from SD mice also suffered colitis phenotype and intestinal microbiota disorder, but no changes in plasma MLT were observed. However, MLT supplementation reversed all inductions in SD-mice, and transplanting feces microbiota from SD + MLT mice significantly restored SD-induced colitis and intestinal microbiota imbalance, including the reduction of *Faecalibactium* and butyrate. Similarly, MLT prevents obesity in mice through modulation of gut microbiota, including increasing the content of *Faecalibactium* [31]. Consequently, our results indicate that MLT effectively improved acute SD-induced colitis via restoring the intestinal microbiota homeostasis especially the increase in *Faecalibactium* and butyrate production.

*Faecalibacterium*, a butyrate producer, can be considered a sensor and a marker of human health [36] because its abundance is associated with intestinal disorders such as IBD, irritable bowel syndrome, and colorectal cancer [37]. Butyrate plays a major role in gut physiology: intrarectal delivery of butyrate-producing bacteria can prevent colitis [19]. In this study, metabolic function and pathway enrichment analysis showed that the decrease in *Faecalibactium* and butyrate content in the colon induced inflammatory response in SD mice. In vivo experiments further showed that acute SD induced weight loss, shorter colon length, fecal occult blood, higher histopathological score as well as increased intestinal permeability and pro-inflammatory factor (TNF-α and IL-17) levels, and decreased Card9 protein and anti-inflammatory factor (IL-10 and IFN-γ) levels. Our results corroborate previous studies in mice that were exposed to three SD-induced typical characteristics of colitis: weight loss, shorter colon length, and fecal occult blood [38]. Genome-wide association studies found strong correlations between loss-of-function Card9 mutations and an increased likelihood of developing inflammatory diseases [39,40]. Specifically, one of the numerous colitis susceptibility factors, Card9-signaling has been implicated in intestinal immune responses and the maintenance of homeostasis after epithelial injury and bacterial infection in mice [41]. Therefore, our studies have confirmed that SD leads to the occurrence of colitis along with a decrease in MLT and butyrate levels.

Interestingly, even though either supplementation with MLT (SD + MLT and SD + Abs + MLT group) or butyrate (SD + Abs + Butyrate group) to SD mice significantly increased butyrate levels and improved SD-induced colitis, the former can increase MLT levels, whereas the latter cannot remove the suppression of MLT secretion, which is the fundamental cause of SD-induced colitis. In particular, a single treatment of mice with ABs showed a reduction in butyrate and the occurrence of colitis, similar to SD treatment, but there was no significant change in MLT levels compared with the control group. MLT suppression plays a crucial role in SD-induced intestinal barrier dysfunction [5]. Consequently, these results suggest that SD inhibits MLT secretion, drives a selective reduction in colonic *Faecalibactium* population and butyrate production, and ultimately induces colitis. Furthermore, the scientific literature has demonstrated that glutamine is one of the main beneficial amino acids. It plays an important role in gut microbiota and immunity [42]. Therefore, the combined supplementation of exogenous MLT and glutamine may be a better choice to treat SD-induced colitis relative to administration of butyrate, although MLT prevents SD-induced colitis via butyrate mediation.

Butyrate exerts anti-inflammatory properties through the downregulation of proinflammatory cytokines [18]. Butyrate may permeate the cell membrane using two different solute transporters, the proton-coupled MCT1 and SMCT1 [20,21], or through the activation of metabolite-sensing G-protein coupled receptors (GPR41, GPR43, and GPR109) [13,22,43]. Once inside the cell, butyrate can regulate cell functions by HDAC inhibition [16,25,26]. This study explored the mechanism of butyrate-mediated MLT-improved SD-induced colitis using in vivo and in vitro tests. In vivo, we found that SD upregulated the expression of HDAC3, p-P65, p-IκB, and NLRP3 and downregulated the expression of MCT1, p-GSK-3β, β-catenin, HIF-1α, and Card9. However, supplementation of MLT or butyrate to SD mice effectively suppressed this process. Similarly, in vitro, butyrate reversed the LPS-induced increase in HDAC3, p-P65, p-IκB, NLRP3, and LDH index and the decrease in MCT1, p-GSK-3β, β-catenin, HIF-1α, Card9, and proliferation activity in IECs. Furthermore, the improvement in butyrate could be blocked by ITSA-1 and Chetomin. However, ITSA-1 did not change the expression of MCT1 and HDAC3, and Chetomin did not affect the expression levels of MCT1, HDAC3, p-GSK-3β, β-catenin, and HIF-1α after butyrate pre-treatment. A previous study confirmed that HDAC3 mediates cardioprotection of remifentanil postconditioning by targeting GSK-3β in H9c2 cardiomyocytes in hypoxia/reoxygenation injury [44]. HIF-1α is involved in the regulation of GSK-3β signaling in H9c2 cells in myocardial I/R injury [45]. Similarly, TWS119 and PDTC mimicked the effect of butyrate in the LPS-treated IECs, including the increase in Card9 and proliferative activity as well as the reduction in p-P65, p-IκB, NLRP3, and LDH index. TWS119 did not affect the expression levels of MCT1, HDAC3, and p-GSK-3β, while PDTC did not affect the expression levels of MCT1, HDAC3, p-GSK-3β, β-catenin, and HIF-1α. Moreover, previous studies revealed physical and functional interactions between HIF and NF-κB [46]. HIF-1α has been shown to restrict NF-κB in vivo and in vitro under inflammatory conditions [46,47]. Our study showed that MLT mediated butyrate-improved SD-induced colitis through the MCT1/HDAC3/P-GSK-3β/HIF-1α/NF-KB loop.

This study provides a deeper understanding of how MLT improved SD-induced colitis by increasing *Faecalibacterium* and its metabolite butyrate production (Figure 7A,B). This finding demonstrates the effectiveness of MLT in the treatment of SD-induced colitis relative to butyrate and may facilitate the development of new treatments using MLT for various gastrointestinal disorders.

## Figures and Tables

**Figure 1 ijms-22-11894-f001:**
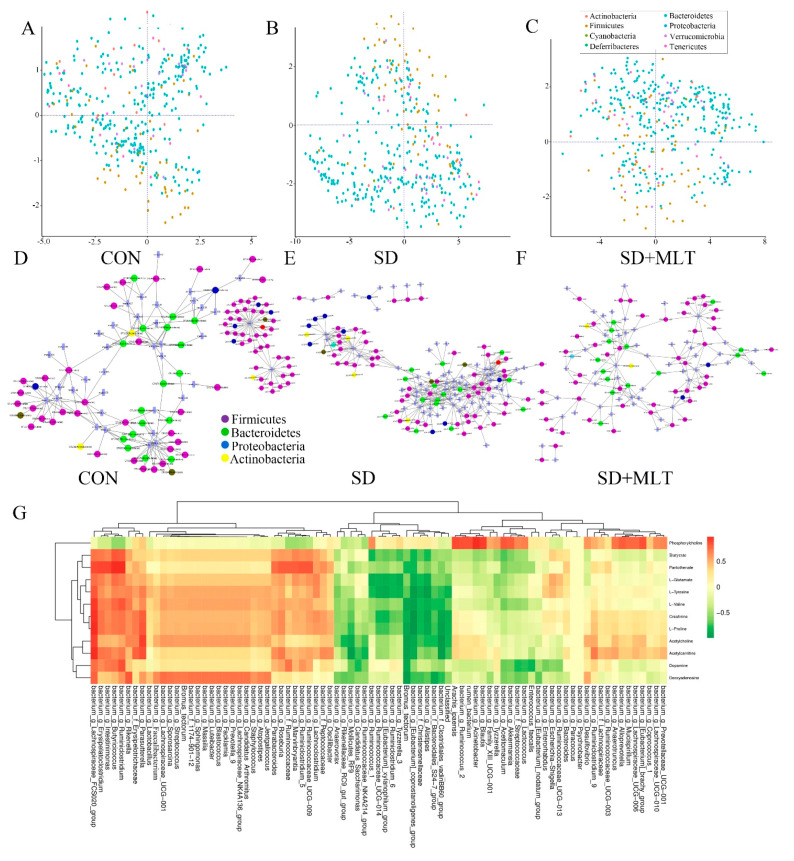
Correlation analysis between intestinal microbiota and metabolites in the colon of SD mice with or without melatonin supplementation (*n* = 8). (**A**–**C**) Co-inertia analysis of differences in microorganisms among the CON (**A**), SD (**B**), and SD + MLT groups (**C**). (**D**–**F**) Correlation network analysis between intestinal microbiota and metabolites among the CON (**D**), SD (**E**), and SD + MLT groups (**F**). (**G**) Correlation of the most significant changes in intestinal microbiota and metabolites among the CON, SD, and SD + MLT groups. CON: non-sleep-deprived control group; SD: sleep deprivation group; SD + MLT: SD + melatonin supplement group.

**Figure 2 ijms-22-11894-f002:**
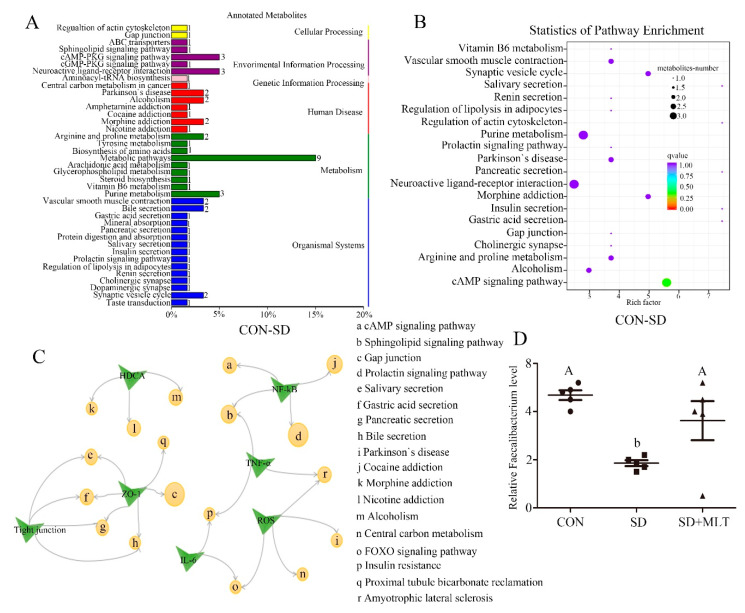
KEGG analysis. KEGG difference analysis (**A**) and KEGG pathway enrichment (**B**), CON group compared to SD group; Links between signal pathways and functions based on KEGG analysis (**C**) (*n* = 8). Relative Faecalibacterium level (**D**) was measured using a metabolomics test in the CON, SD, and SD + MLT groups. Values are presented as the mean ± SE. Differences were assessed using ANOVA and are denoted as follows: different lowercase letters or different uppercase letters: *p* < 0.05; different letters and one contains uppercase letters and the other is lowercase letters: *p* < 0.01; same letter: *p* > 0.05.

**Figure 3 ijms-22-11894-f003:**
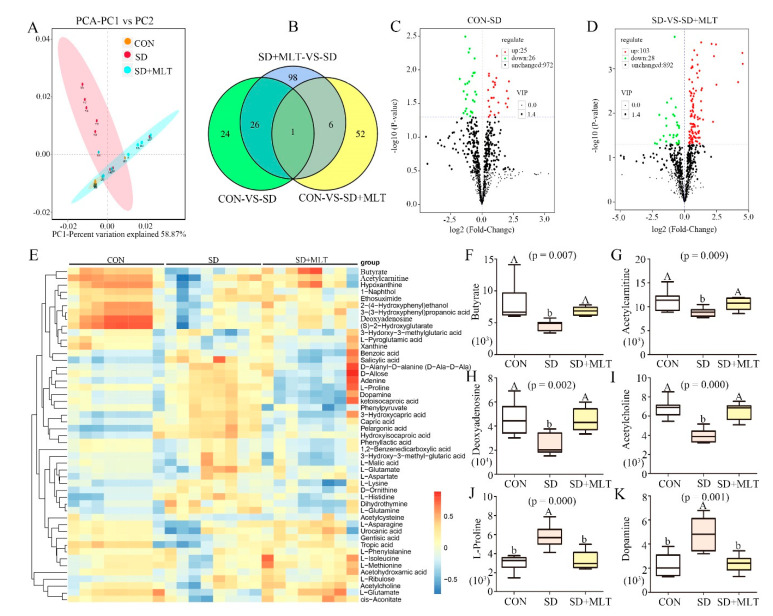
Composition of the colon microbiota metabolites in SD mice with or without melatonin supplementation (*n* = 8). The β-diversity of principal component analysis (PCA) (**A**) and Venn based on the microbiota metabolites (**B**) in the CON, SD, and SD + MLT groups; volcano plot based on the differential metabolite screening (**C**), compared with the CON and SD groups; volcano plot based on the differential metabolite screening (**D**), compared with the SD and SD + MLT groups; heatmap showing the relative abundance of the key identified 49 metabolites that were significantly altered by melatonin in SD mice (**E**). The relative abundance of butyrate (**F**), acetylcarnitine (**G**), deoxyadenosine (**H**), acetylcholine (**I**), L-proline (**J**), and dopamine (**K**) in the CON, SD, and SD + MLT groups in the colon microbiota based on the heatmap results. CON: non-sleep-deprived control group; SD: sleep deprivation group; SD + MLT: SD + melatonin supplement group. Differences were assessed using ANOVA and are denoted as follows: different lowercase letters or different uppercase letters: *p* < 0.05; different letters and one contains uppercase letters and the other is lowercase letters: *p* < 0.01; same letter: *p* > 0.05.

**Figure 4 ijms-22-11894-f004:**
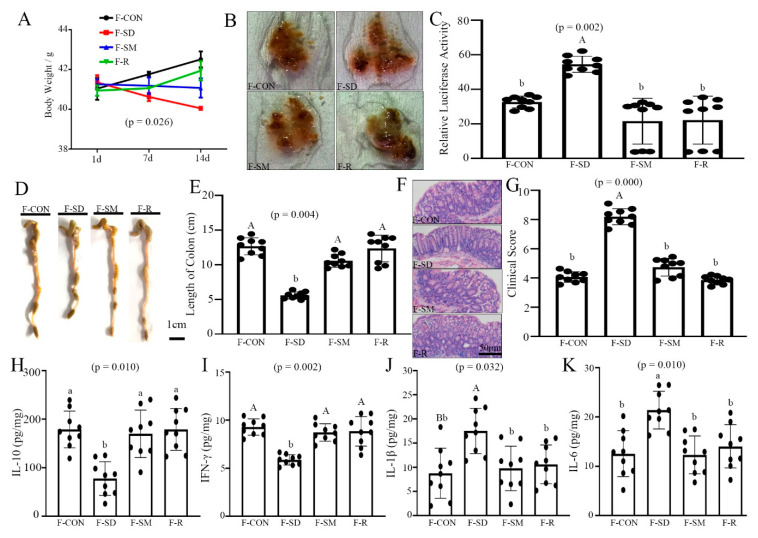
FMT re-established the intestinal microecology similar to CON, SD, and SD + MLT mice. (**A**) Body weight; (**B**) fecal occult blood; (**C**) relative luciferase activity for colonic permeability; (**D**), (**E**) colonic length; (**F**) H&E staining photographs (scale: 50 μm); (**G**) histopathological score; (**H**–**K**): IL-10 (**H**), IFN-γ (**I**), IL-1β (**J**), IL-6 (**K**) concentrations in the colon of the F-CON, F-SD, F-SM, and F-R groups. Values are presented as the mean ± SE. Differences were assessed using ANOVA and are denoted as follows: different lowercase letters or different uppercase letters: *p* < 0.05; different letters and one contains uppercase letters and the other is lowercase letters: *p* < 0.01; same letter: *p* > 0.05.

**Figure 5 ijms-22-11894-f005:**
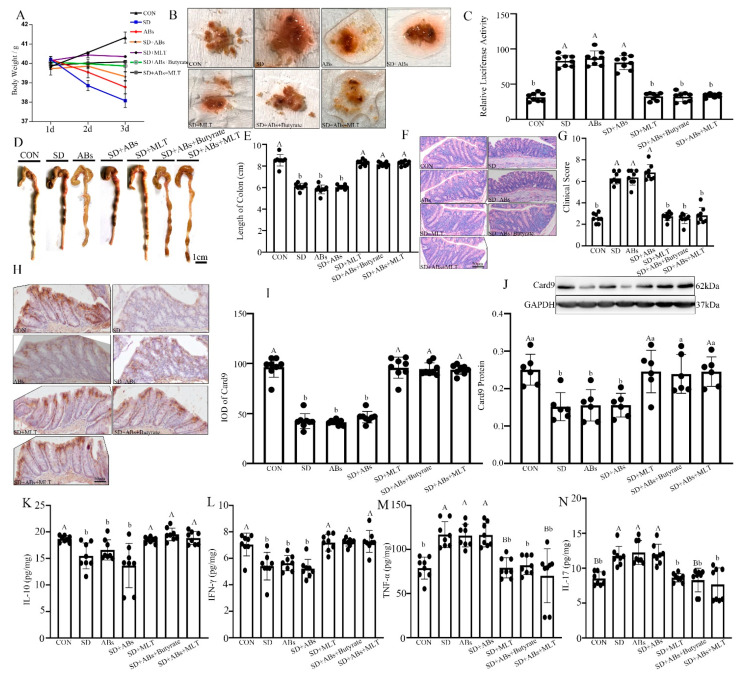
Effect of melatonin and butyrate supplementation on improving SD-induced colitis in mice. (**A**) Body weight; (**B**) fecal occult blood; (**C**) relative luciferase activity for colonic permeability; (**D**,**E**) colonic length; (**E**,**F**,**H**) staining photographs (scale: 50 μm); (**G**) histopathological score; (**H**) immunohistochemical staining photographs of Card9 (scale: 50 μm); (**I**) integral optical density (IOD) of Card9-positive cells; (**J**) relative protein level of Card9; (**K**–**N**): IL-10 (**K**), IFN-γ (**L**), TNF-α (**M**), IL-17 (**N**) concentrations in the colon of the CON, SD, ABs, Abs + SD, SD + MLT, Abs + SD + Butyrate, and Abs + SD + MLT groups (*n* = 9). Values are presented as the mean ± SE. Differences were assessed using ANOVA and are denoted as follows: different lowercase letters or different uppercase letters: *p* < 0.05; different letters and one contains uppercase letters and the other is lowercase letters: *p* < 0.01; same letter: *p* > 0.05.

**Figure 6 ijms-22-11894-f006:**
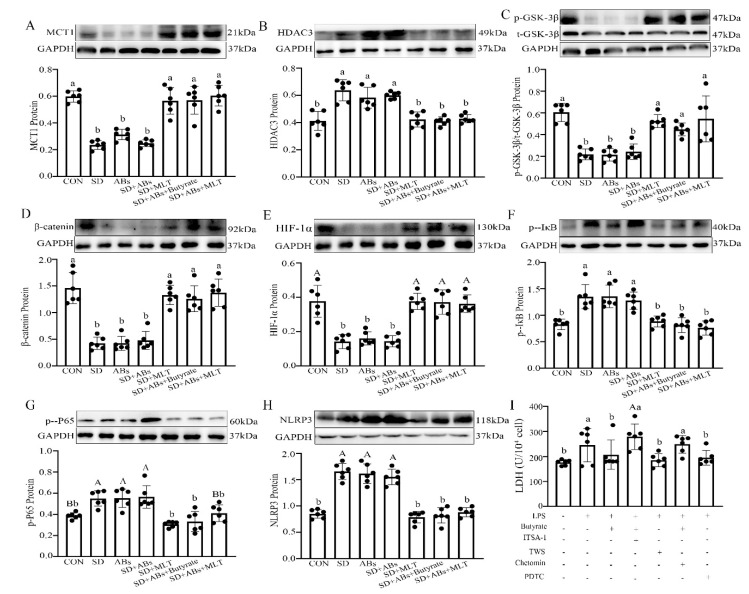
Effects of melatonin and butyrate supplementation on the relative levels of MCT1 (**A**), HDAC3 (**B**), p-GSK-3β (**C**), β-catenin (**D**), HIF-1α (**E**), p-IκB (**F**), p-P65 (**G**), and NLRP3 (**H**) in the colon of the CON, SD, ABs, Abs + SD, SD + MLT, Abs + SD + Butyrate, and Abs + SD + MLT groups (*n* = 12). LDH index in LPS-treated IECs (**I**). Chetomin, an antagonist of HIF-1α; PDTC, an antagonist of NF-κB; ITSA-1, an agonist of HDAC3; TWS, an antagonist of GSK-3β. Values are presented as the mean ± SE. Differences were assessed using ANOVA and are denoted as follows: different lowercase letters or different uppercase letters: *p* < 0.05; different letters and one contains uppercase letters and the other is lowercase letters: *p* < 0.01; same letter: *p* > 0.05.

**Figure 7 ijms-22-11894-f007:**
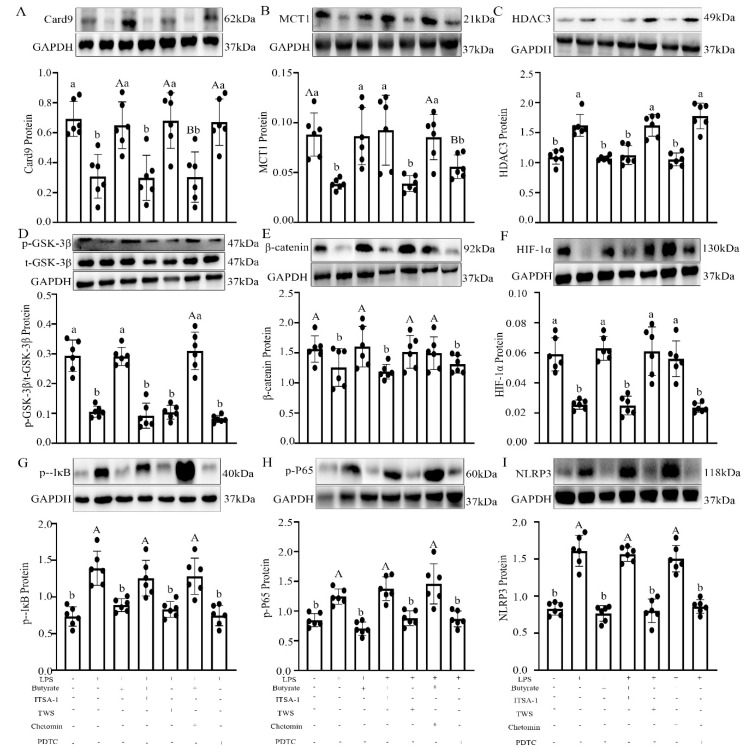
Improvement effect of butyrate on the relative levels of Card9 (**A**), MCT1 (**B**), HDAC3 (**C**), p-GSK-3β (**D**), β-catenin (**E**), HIF-1α (**F**), p-IκB (**G**), p-P65 (**H**), and NLRP3 (**I**) in LPS-treated IECs. Chetomin, an antagonist of HIF-1α; PDTC, an antagonist of NF-κB; ITSA-1, an agonist of HDAC3; TWS, an antagonist of GSK-3β. Values are presented as the mean ± SE. Differences were assessed using ANOVA and are denoted as follows: different lowercase letters or different uppercase letters: *p* < 0.05; different letters and one contains uppercase letters and the other is lowercase letters: *p* < 0.01; same letter: *p* > 0.05.

**Table 1 ijms-22-11894-t001:** Changes in plasma melatonin level and colonic SCFAs production in mice.

Group	MLT (pg/mL)	Butyrate (µg/mg)	Acetate (µg/mg)	Propiornate (µg/mg)	Colitis
CON	24,086 ± 0.236 ^a^	3.252 ± 0.009 ^Aa^	1.000 ± 0.497	0.789 ± 0.557	−
SD	20,431 ± 0.214 ^b^	2.587 ± 0.020 ^b^	1.146 ± 0.399	0.904 ± 0.447	+
ABs	23,798 ± 0.223 ^a^	2.584 ± 0.004 ^b^	0.989 ± 0.204	0.781 ± 0.229	+
SD + ABs	19,879 ± 0.593 ^b^	2.540 ± 0.013 ^b^	1.106 ± 0.267	0.873 ± 0.299	+
SD + MLT	23,955 ± 0.431 ^a^	3.176 ± 0.019 ^a^	1.113 ± 0.131	0.878 ± 0.147	−
SD + Abs + MLT	24,363 ± 0.398 ^a^	3.202 ± 0.018 ^Aa^	1.072 ± 0.049	0.846 ± 0.054	−
SD + Abs + Butyrate	20,102 ± 0.225 ^b^	3.682 ± 0.033 ^Aa^	1.011 ± 0.094	0.797 ± 0.105	−

Improving effect of melatonin on the content of plasma melatonin, colonic butyrate, colonic acetate and colonic propionate in sleep-deprived mice (*n* = 8). Values are presented as the means ± SE. Differences were assessed by ANOVA and denoted as follows: different lowercase letters or different uppercase letters: *p* < 0.05; different letters and one contains uppercase letters and the other is lowercase letters: *p* < 0.01; same letter: *p* > 0.05. CON: non-sleep-deprived control group; SD: sleep deprivation group; ABs: only antibiotics supplementation; SD + ABs: SD + antibiotics; SD + MLT: SD + melatonin supplementation; SD + ABs + MLT: SD + antibiotics + melatonin supplementation; SD + ABs + Butyrate: SD + antibiotics + butyrate supplementation.

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
