# Peer review of "Melatonin-Mediated Colonic Microbiota Metabolite Butyrate Prevents Acute Sleep Deprivation-Induced Colitis in Mice"

_ijms, 2021, doi:10.3390/ijms222111894_

Round 1
Reviewer 1 Report
This study explored the mechanism 15 whereby exogenous melatonin prevented SD-induced colitis.
The article touch a very hard topic, with an high potential level of citations and scientific impact.
I have really few comments, below the main comment
I strongly suggest to discuss the effect on the axis glutamine and gut microbiota and the possible effect on melatonin in combination with glutamine following the article of
Perna S, Alalwan TA, Alaali Z, Alnashaba T, Gasparri C, Infantino V, Hammad L, Riva A, Petrangolini G, Allegrini P, Rondanelli M. The Role of Glutamine in the Complex Interaction between Gut Microbiota and Health: A Narrative Review. Int J Mol Sci. 2019 Oct 22;20(20):5232. doi: 10.3390/ijms20205232. PMID: 31652531; PMCID: PMC6834172.Author Response
Please see the attachment.

Reviewer 2 Report
I have read with an interest this original paper of Gao et al. The authors took up an interesting topic of the melatonin-mediated colonic microbiota metabolite butyrate in the context of acute sleep deprivation-induced
colitis in mice. In my opinion, the following issues should be addressed:
Major:
1) As the introduction is pertaining to IBD in the context of sleep disorders, the information about sleep quality among IBD patients should be added (e.g, 10.3390/jcm9092921, 10.1111/nmo.13978)
2) The Statistical Analysis should be described in the separated section. Did you check the normality of obtained data? Average +- SD should be used for data with normal distribution, median with IQR should be employed in other cases. Please state p<0.001 instead of p=0.000.
Minor:
1) The titles of subsections should be removed from the abstract (e.g., background, results)
2) Strengths and limitations of the study should be listed in the separated section.
Author Response
Reviewers' comments:
Comments and Suggestions for Authors
I have read with an interest this original paper of Gao et al. The authors took up an interesting topic of the melatonin-mediated colonic microbiota metabolite butyrate in the context of acute sleep deprivation-induced colitis in mice. In my opinion, the following issues should be addressed:
- As the introduction is pertaining to IBD in the context of sleep disorders, the information about sleep quality among IBD patients should be added (e.g, 10.3390/jcm9092921, 10.1111/nmo.13978)
Answer: Thanks you for your suggestion and question. I have discussed this and cited this document. Please see page 2 lines 45-47 and page 18 lines 515-519 in the revised manuscript (R1).
Reference
Sochal M, Małecka-Panas E, Gabryelska A, Talar-Wojnarowska R, Szmyd B, Krzywdzińska M, Białasiewicz P. Determinants of sleep quality in inflammatory bowel diseases. J Clin Med. 2020;9:2921.
Sochal M, Małecka-Panas E, Gabryelska A, Talar-Wojnarowska R, Szmyd B, Białasiewicz P. Brain-derived neurotrophic factor is elevated in the blood serum of Crohn's disease patients, but is not influenced by anti-TNF-α treatment-A pilot study. Neurogastroenterol Motil. 2021;33:e13978.
The Statistical Analysis should be described in the separated section. Did you check the normality of obtained data? Average +- SD should be used for data with normal distribution, median with IQR should be employed in other cases. Please state p<0.001 instead of p=0.000.
Answer: Thanks you for your suggestion and question. We have modified and supplemented the results analysis description in the materials and methods, and added the standard deviation in the results. Please see page 5 lines 169-173 and pages 5-13 lines 176-391 in the revised manuscript (R1).
- The titles of subsections should be removed from the abstract (e.g., background, results)
Answer: Thanks you for your suggestion. We have removed the titles of subsections from the abstract.
- Strengths and limitations of the study should be listed in the separated section.
Answer: Thanks you for your suggestion. Strengths and limitations are as follows:
Strengths:
- SD led to a decrease in plasma MT levels, which decreased Faecalibacterium and butyrate production in the colon.
- MT can prevent SD-induced colitis by increasing the Faecalibacterium population and butyrate production.
- Administration of butyrate can alleviate SD-induced colitis, but cannot remove the suppression of MT secretion
- The reduction of MT is the fundamental cause of SD-induced colitis.
Limitations:
- The mechanism of sleep deprivation-induced reduction of melatonin levels needs further exploration.
- The mechanism of melatonin-induced increase in the relative abundance of Faecalibacterium needs to be further explored.
We have already added it. Please see the file named “Strengths and limitations”.

Round 2
Reviewer 2 Report
The authors addressed all the comments sufficiently improving the manuscript.
This manuscript is a resubmission of an earlier submission. The following is a list of the peer review reports and author responses from that submission.
Round 1
Reviewer 1 Report
the paper is well written and deserves the publication.
My main concern is related to the "translational" aspect of this research on clinical practice.
Author Response
Answer: Thanks you for your suggestion. This is also the issue we will consider in the future.
Reviewer 2 Report
The authors previously published their findings on sleep deprivation and melatonin supplementation on gut inflammation and colon microbiota disorders. Here they extend their research by looking at impacts on short-chain fatty acids like butyrate that are known to participate in the crosstalk between the microbiota and host immune system. The authors explore the impact of FMT from mice -/+ SD treated with different experimental conditions to explore the roles of melatonin and SCFAs. The findings are clear and significant but whether the observed effects are due to melatonin itself or the impact of melatonin on sleep and the biochemical factors that are perturbed in the presence or absence of sleep deprivation remains unclear. What is the impact of MLT in absence of SD? Is it a biochemical response to MLT itself or is MLT abolishing SD and therefore impacting a number of biochemical factors that result in control phenotype?
While the science is fine, overall this paper was carelessly put together and needs significant edits and modifications before it can be published.
See below:
The first sentence in the abstract – revise for English and scientific style
Abstract, in general, is written as if copied and pasted from another format. Please edit and revise for readability and format. Example – line 15: background radical, line 16: methods 16, line 19: coli-tis, line 27: re-versed
Figure 1 – hard to read due to resolution, cannot assess
Figure 2 – b, ven diagram labeling is unclear and numbers in text don’t match numbers in figure, SD MLT volcano plot should be in main figure, not supplemental
Page 7 line 248- “However, MLT supplementation reversed these changes to the control levels (p > 0.55).” – is this p-value saying that the difference between control and MLT is not significantly different, or that the difference between SD and MLT treated is not significantly different – please clarify.
2E-J – add p values to figures
Fig 3 is cut off
Fig 4 – p-value annotation is hard to follow, why not just note p values on the figure?
I am confused by figures S1 and S2 - I don’t see a mention of them in the text or an accompanying legend for any of the supplemental figs
line 344 – butyrate spelling error
Figure 5 does not seem to match the legend
Fig 7 doesn’t match the legend
Author Response
Answer to the Reviewers' Comments (ijms-1307230)
Dear Prof. Paschalis Alexandridis and Reviewers,
We delivered the article for the first time on 5-July-2021, and received a reply on 11-August-2021. I’m very glad to hear from you, and thank you for your works in dealing with my paper (ijms-1307230). All of those comments are valuable and very helpful for improving the quality of our paper, as well as the important guiding to our research. We have carefully checked the manuscript and revised it according to the suggestions [red font color represented the revised text in revised manuscript]. Now, I’m submitting here with a reply of queries.
Sincerely yours,
Yaoxing Chen, Prof., Ph.D.
Beijing Advanced Innovation Center for Food Nutrition and Human Health,
College of Animal Medicine, China Agricultural University
Haidian, Beijing, 100193, China
- mail: yxchen@cau.edu.cn
Reviewer
Comments and Suggestions for Authors
The authors previously published their findings on sleep deprivation and melatonin supplementation on gut inflammation and colon microbiota disorders. Here they extend their research by looking at impacts on short-chain fatty acids like butyrate that are known to participate in the crosstalk between the microbiota and host immune system. The authors explore the impact of FMT from mice -/+ SD treated with different experimental conditions to explore the roles of melatonin and SCFAs. The findings are clear and significant but whether the observed effects are due to melatonin itself or the impact of melatonin on sleep and the biochemical factors that are perturbed in the presence or absence of sleep deprivation remains unclear. What is the impact of MLT in absence of SD? Is it a biochemical response to MLT itself or is MLT abolishing SD and therefore impacting a number of biochemical factors that result in control phenotype?
Answer: Thanks you for your suggestion and question. The improvement effect of melatonin on sleep deprivation-induced colitis is that melatonin itself plays a role, not that it regulates sleep. Because the mice have been in a state of sleep deprivation during the entire experiment and did not sleep. In addition, the only variable between the control group and the sleep deprivation group is sleep, and other conditions are the same. Therefore, there are no other biochemical factors that will affect the results of the experiment. Specifically, after sleep deprivation, the disturbance of intestinal microbiota including the decline of Faecalibacterium and its metabolites butyrate, induced the occurrence of colitis in mice. However, melatonin supplementation can restore the homeostasis of the intestinal microbiota by increasing the abundance of Faecalibacterium and butyrate and ultimately improve sleep deprivation-induced colitis. Is it a biochemical response to melatonin itself and therefore impacting a number of biochemical factors that result in control phenotype, instead of melatonin abolishing sleep deprivation.
- The first sentence in the abstract – revise for English and scientific style
Abstract, in general, is written as if copied and pasted from another format. Please edit and revise for readability and format. Example – line 15: background radical, line 16: methods 16, line 19: coli-tis, line 27: re-versed.
Answer: Thanks you for your suggestion and question. We have re-edited and revised the abstract for readability and format. Please see page 1 lines 12-32 in the revised manuscript (R1).
- Figure 1 – hard to read due to resolution, cannot assess
Answer: Thanks you for your suggestion and question. We have modified the figure 1. Moreover, we added the files (table S1-S3) corresponding to figure 1D-F. Please see figure 1 and table S1-S3 in the revised manuscript (R1).
- Figure 2 – b, ven diagram labeling is unclear and numbers in text don’t match numbers in figure, SD MLT volcano plot should be in main figure, not supplemental
Answer: Thanks you for your suggestion and question. We have modified the ven diagram labeling. The value in the left circle represents the number of OTUs changed after sleep deprivation compared with the control group, the value in the middle circle represents the number of OTUs changed after the melatonin supplementation compared with the sleep deprivation group, and the intersection of the two circles indicates the amount of OTU recovered to the control group after the melatonin supplementation. Therefore, the numbers in text match numbers in figure. In addition, we have added the SD+MLT volcano plot in main figure. We have modified the figure 1. Please see figure 2 in the revised manuscript (R1).
- Page 7 line 248- “However, MLT supplementation reversed these changes to the control levels (p > 0.55).” – is this p-value saying that the difference between control and MLT is not significantly different, or that the difference between SD and MLT treated is not significantly different – please clarify.
Answer: Thanks you for your suggestion and question. The p-value indicated that the difference between control and MLT is not significantly different. We have modified the sentence. Please see page 7 line 238-239 in the revised manuscript (R1).
- 2E-J – add p values to figures
Answer: Thanks you for your suggestion. We have added the p values to figures. Please see Figure 2E-J in the revised manuscript (R1).
- Fig 3 is cut off.
Answer: Thanks you for your suggestion. We still want to keep figure 3. Figure3A and figure 3B are intestinal metabolic pathways that may be affected by the metabolites predicted by sequencing. Figure 3C is a metabolic pathway that we have screened out and is highly consistent with in vivo experiments, which are closely related the mechanism of butyrate. In addition, figure 3D is the relative abundance of Faecalibacterium in the intestines of the control group, sleep deprivation group, and sleep deprivation group supplemented with melatonin, which is significantly positively correlated with the butyrate content and is vital.
- Fig 4 – p-value annotation is hard to follow, why not just note p values on the figure?
Answer: Thanks you for your suggestion. We have added the p values to figures. Please see Figure 4 in the revised manuscript (R1).
- I am confused by figures S1 and S2 - I don’t see a mention of them in the text or an accompanying legend for any of the supplemental figs
Answer: Thanks you for your suggestion and question. The description of figure S1 and S2 in the Supplementary Materials. Please see page 1 lines 12-13 in the revised Supplementary Materials (R1).
- line 344 – butyrate spelling error
Answer: Thanks you for your suggestion. We have modified the spelling error. Please see lines 344 in the revised manuscript (R1).
- Figure 5 does not seem to match the legend
Answer: Thanks you for your suggestion. We have modified the legend. Please see page 12-13 lines 349-356 in the revised manuscript (R1).
- Fig 7 doesn’t match the legend
Answer: Thanks you for your suggestion. We have modified the legend. Please see page 15 lines 413-418 in the revised manuscript (R1).

Reviewer 3 Report
The authors demonstrated the anti-inflammatory effects of exogenous melatonin on sleep deprivation colitis via positive changes in gut microbiota ( increase of Faecalibacterium population and their metabolite butyrate.
What are the mechanisms of colitis in sleep deprivation model ?? This should be better explained.
Moreover, melatonin improved gut microbiota metabolite disproportionality in SD mice.
What is the effect of melatonin on antimicrobial peptide (AMP) ??
Melatonin has a strong antioxidative effect. Did authors investigated this aspect ? This topic should be discussed.
Is there any evidence that melatonin has positive effects on colits in human?? Please discuss it in term of translatory medicine.
Author Response
Answer to the Reviewers' Comments (ijms-1307230)
Dear Prof. Paschalis Alexandridis and Reviewers,
We delivered the article for the first time on 5-July-2021, and received a reply on 11-August-2021. I’m very glad to hear from you, and thank you for your works in dealing with my paper (ijms-1307230). All of those comments are valuable and very helpful for improving the quality of our paper, as well as the important guiding to our research. We have carefully checked the manuscript and revised it according to the suggestions [red font color represented the revised text in revised manuscript]. Now, I’m submitting here with a reply of queries.
Sincerely yours,
Yaoxing Chen, Prof., Ph.D.
Beijing Advanced Innovation Center for Food Nutrition and Human Health,
College of Animal Medicine, China Agricultural University
Haidian, Beijing, 100193, China
- mail: yxchen@cau.edu.cn
Reviewers' comments:
Comments and Suggestions for Authors
The authors demonstrated the anti-inflammatory effects of exogenous melatonin on sleep deprivation colitis via positive changes in gut microbiota (increase of Faecalibacterium population and their metabolite butyrate.
- What are the mechanisms of colitis in sleep deprivation model? This should be better explained. Moreover, melatonin improved gut microbiota metabolite disproportionality in SD mice.
Answer: Thanks you for your suggestion and question. Our previous studies have found that after sleep deprivation, the melatonin level secreted by the pineal gland of mice was significantly reduced (Gao et al., 2019), which in turn activated the HPA axis, leading to an intestinal oxidative stress and further inducing intestinal microbiota imbalance (Gao et al., 2021). Our results in this study further showed that after sleep deprivation, the abundance of Faecalibacterium and its metabolite butyrate decreased significantly, which in turn reduced the expression of butyrate receptor MCT1 and activates HDAC3/GSK-3β/β-catenin /p-HIF-1α/NF-κB/NLRP3 pathway, as well as up-regulated the levels of pro-inflammatory factors (IL-10 and IFN-γ) and down-regulated the levels of anti-inflammatory factors (IL-17 and TNF-α), ultimately inducing the occurrence of colitis. The effect of melatonin supplementation on the intestinal microbiota is disproportionality, which may be related to the mechanism of melatonin and the characteristics of the microbiota itself. We will consider this in future research.
References
Gao, T., Wang, Z., Dong, Y., Cao, J., Lin, R., Wang, X., Yu, Z., Chen, Y. Role of melatonin in sleep deprivation-induced intestinal barrierdysfunction in mice. J Pineal Res. 2019;67:e12574.
Gao, T., Wang, Z., Cao, J., Dong, Y., Chen, Y. Melatonin ameliorates corticosterone-mediated oxidative stress induced colitis in sleep-deprived mice involving in gut microbiota. Oxid Med Cell Longev. 2021;2021:9981480.
- What is the effect of melatonin on antimicrobial peptide (AMP) ?
Answer: Thanks you for your suggestion and question. Antimicrobial peptides are synthesized and secreted by Paneth cells in the small intestines. We did not detect the expression of antimicrobial peptides in the intestines of each group of mice, but we did detect the number of Paneth cells. The results showed that after sleep deprivation, the number of Paneth cells in the small intestine was significantly reduced, while melatonin supplementation could reverse this process (data have not shown).
- Melatonin has a strong antioxidative effect. Did authors investigate this aspect? This topic should be discussed.
Answer: Thanks you for your suggestion and question. Melatonin is mainly secreted by the pineal gland and is one of the most well-investigated antioxidants. It can scavenge a variety of free radicals (Manchester et al., 2015; Reiter et al., 2016), upregulate the expression of antioxidant proteins (Zhang and Zhang, 2014; Wang et al., 2012) and defend against oxidants-induced damage in many tissues (Gao et al., 2016). Therefore, melatonin can modulate the antioxidant defense system by increasing the activities of antioxidant enzymes, and to stimulate the innate immune response through its direct and indirect actions (Esteban-Zubero et al., 2017). Our previous research showed that, in dextran sodium sulfate-induced colitis, melatonin-mediated MT2 activated PI3K/AKT/Nrf2/RORα/SIRT1 pathway and suppressed NF-κB pathway, ultimately improved DSS-induced colitis, which provides evidence for melatonin as an efficient therapy against oxidative stress associated IBD (Gao et al., 2021).
References
Esteban-Zubero, E.; López-Pingarrón, L.; Alatorre-Jiménez, M.A.; Ochoa-Moneo, P.; Buisac-Ramón, C.; Rivas-Jiménez, M.; Castán-Ruiz, S.; Antoñanzas-Lombarte, Á.; Tan, D.X.; García, J.J.; Reiter, RJ. Melatonin's role as a co-adjuvant treatment in colonic diseases: A review. Life Sci. 2017,170, 72-81.
Gao, L.; Zhao, Y.C.; Liang, Y.; Lin, X.H.; Tan, YJ.; Wu, D.D.; Li, X.Z.; Ye, B.Z.; Kong, F.Q.; Sheng, J.Z.; Huang, H.F. The impaired myocardial ischemic tolerance in adult offspring of diabetic pregnancy is restored by maternal melatonin treatment. J Pineal Res. 2016, 61, 340-352.
Gao, T.; Wang, T.; Wang, Z.; Cao, J.; Dong, Y.; Chen, Y. Melatonin-mediated MT2 attenuates colitis induced by dextran sodium sulfate via PI3K/AKT/Nrf2/SIRT1/RORα/NF-κB signaling pathways. Int Immunopharmacol. 2021, 96, 107779.
Manchester, L.C.; Coto-Montes, A.; Boga, J.A.; Andersen, L.P.H.; Zhou, Z.; Galano, A.; Vriend J.; Tan, D.X.; Reiter, R.J.; Melatonin: an ancient molecule that makes oxygen metabolically tolerable. J Pineal Res. 2015, 59, 403-419
Reiter, R.J.; Mayo, J.C.; Tan, D.X.; Sainz, R.M.; Alatorre-Jimenez, M.; Qin, L. Melatonin as an antioxidant: under promises but over delivers. J Pineal Res. 2016, 61, 253-278.
Wang, Z.; Ma, C.; Meng, C.J.; Zhu, G.Q.; Sun, X.B.; Huo, L.; Zhang, J.; Liu H.X.; He, W.C.; Shen, X.M.; Shu, Z.; Chen, G. Melatonin activates the Nrf2-ARE pathway when it protects against early brain injury in a subarachnoid hemorrhage model. J Pineal Res. 2012, 53, 129-137.
Zhang, H.M.; Zhang, Y. Melatonin: a well-documented antioxidant with conditional pro-oxidant actions. J Pineal Res. 2014, 57, 131-146.
- Is there any evidence that melatonin has positive effects on colitis in human?? Please discuss it in term of translatory medicine.
Answer: Thanks you for your suggestion and question. Melatonin is a major hormone molecular product released from the pineal gland into the circulatory system to regulate the immune system and circadian rhythms (Auld et al., 2017). It has been studied as a co-adjuvant treatment in several gastrointestinal diseases including irritable bowel syndrome (IBS), constipation-predominant IBS, diarrhea-predominant IBS, Crohn's disease, ulcerative colitis, and necrotizing enterocolitis (Esteban-Zubero et al., 2017). Serving as a multitasking molecule and primary signal mediating microbial metabolism, circadian rhythms and intestinal mucosal immune cells, melatonin has the potential to attend therapy on intestinal diseases in a substantial way. Specifically, intestinal melatonin possesses immune regulatory properties in the gut as a fundamental component of neuro-immuno-modulation (Chen et al., 2017; Huang et al., 2013), such as acting as a regulator of intestinal inflammation, serving as an appropriate player in the immune system or working as a natural antioxidant for immune stimulation. These modulatory actions are carried out in different ways, including the amphophilic properties that are resistant to free radical attacks and the regulation and modification of various cells and cytokines in the immune system.
Considering that melatonin is both aqueous and lipid soluble and can pass through all cell barriers, including cell membranes and blood‐brain barriers (Bagga et al., 2018; Venegas et al., 2012), melatonin is possibly an endocrine transmission signal from commensal microbes and may serve as a “meta‐organism” responding to the host (Leone et al., 2015). Through these means, melatonin can affect the molecular clock of microbes, regulate the rhythm of microbial metabolites (Kim et al., 2016), and also affect the host immune function (da Silveira Cruz‐Machado et al., 2017). Therefore, targeting melatonin in identified microbial metabolic pathways and then altering the rhythm and output of metabolites to improve host’s gut health is a potential candidate option for future intestinal disease treatment.
The mechanism of melatonin improving colitis is closely related to the cause of colitis. In this study, administration MLT may be a better therapy for SD-induced colitis relative to butyrate. A feasible mechanism would involve that melatonin up-regulated the Faecalibacterium population and its metabolite butyrate production and MCT1 expression and inhibited HDAC3 in colon, which would allow p-GSK-3β/β-catenin/HIF-1α activation and NF-κB/NLRP3 suppression to up-regulate Card9 expression and suppress inflammation response.
Reference
Auld, F.; Maschauer, E.L.; Morrison, I.; Skene, D.J.; Riha, R.L. Evidence for the efficacy of melatonin in the treatment of primary adult sleep disorders. Sleep Med Rev. 2017, 34, 10‐22.
Bagga, D.; Reichert, J.L.; Koschutnig, K.; Aigner, C.S.; Holzer, P.; Koskinen, K.; Moissl-Eichinger, C.; Schöpf, V. Probiotics drive gut microbiome triggering emotional brain signatures. Gut Microbes. 2018, 9, 486-496.
Chen, X.; Eslamfam, S.; Fang, L.; Qiao, S.; Ma, X. Maintenance of gastrointestinal glucose homeostasis by the gut‐brain axis. Curr Protein Pept Sc. 2017, 18, 541‐547.
da Silveira Cruz‐Machado, S.; Tamura, E.K.; Carvalho‐Sousa, C.E.; Rocha, V.A.; Pinato, L.; Fernandes, P.A.C.; Markus, R.P. Daily corticosterone rhythm modulates pineal function through NFκB‐related gene transcriptional program. Sci Rep. 2017, 7, 2091.
Esteban-Zubero, E.; López-Pingarrón, L.; Alatorre-Jiménez, M.A.; Ochoa-Moneo, P.; Buisac-Ramón, C.; Rivas-Jiménez, M.; Castán-Ruiz, S.; Antoñanzas-Lombarte, Á.; Tan, D.X.; García, J.J.; Reiter, R.J. Melatonin's role as a co-adjuvant treatment in colonic diseases: A review. Life Sci. 2017, 1, 170:72-81.
Huang, H.; Wang, Z.; Weng, S.J.; Sun, X.H.; Yang, X.L. Neuromodulatory role of melatonin in retinal information processing. Prog Retin Eye Res. 2013, 32, 64‐87.
Kim, M.; Qie, Y.; Park, J.; Kim, C.H. Gut microbial metabolites fuel host antibody responses. Cell Host Microbe. 2016, 20, 202‐214.
Leone, V.; Gibbons, S.M.; Martinez, K.; Hutchison, A.L.; Huang, E.Y.; Cham, C.M.; Pierre, J.F.; Heneghan, A.F.; Nadimpalli, A.; Hubert, N.; Zale, E.; Wang, Y.; Huang, Y.; Theriault, B.; Dinner, A.R.; Musch, M.W.; Kudsk, K.A.; Prendergast, B.J.; Gilbert, J.A.; Chang, E.B. Effects of diurnal variation of gut microbes and high‐fat feeding on host circadian clock function and metabolism. Cell Host Microbe. 2015, 17, 681‐689.
López-Pingarrón, L.; Alatorre-Jiménez, M.A.; Ochoa-Moneo, P.; Buisac-Ramón, C.; Rivas-Jiménez, M.; Castán-Ruiz, S.; Antoñanzas-Lombarte, Á.; Tan, D.X.; García, J.J.; Reiter, R.J. Melatonin's role as a co-adjuvant treatment in colonic diseases: A review. Life Sci. 2017, 170, 72-81.
Venegas, C.; García, J.A.; Escames, G.; Ortiz, F.; López, A.; Doerrier, C.; García-Corzo, L.; López, L.C.; Reiter, R.J.; Acuña-Castroviejo, D.. Extrapineal melatonin: analysis of its subcellular distribution and daily fluctuations. J Pineal Res. 2012, 52,217‐227.

Round 2
Reviewer 2 Report
Improved. Comments addressed.